# Effect of Clemastine on Neurophysiological Outcomes in an Ovine Model of Neonatal Hypoxic-Ischemic Encephalopathy

**DOI:** 10.3390/children10111728

**Published:** 2023-10-25

**Authors:** Jana Krystofova Mike, Yasmine White, Rachel S. Hutchings, Christian Vento, Janica Ha, Ariana Iranmahboub, Hadiya Manzoor, Anya Gunewardena, Cheryl Cheah, Aijun Wang, Brian D. Goudy, Satyan Lakshminrusimha, Janel Long-Boyle, Jeffrey R. Fineman, Donna M. Ferriero, Emin Maltepe

**Affiliations:** 1Department of Pediatrics, University of California San Francisco, San Francisco, CA 94158, USArachel.hutchings@ucsf.edu (R.S.H.); ariana.iranmahboub@ucsf.edu (A.I.); cheryl.cheah@ucsf.edu (C.C.); donna.ferriero@ucsf.edu (D.M.F.);; 2Department of Biomedical Engineering, University of California Davis, Davis, CA 95817, USA; aawang@ucdavis.edu; 3Department of Pediatrics, University of California Davis, Davis, CA 95817, USAslakshmi@ucdavis.edu (S.L.); 4School of Pharmacy, University of California San Francisco, San Francisco, CA 94143, USA; 5Initiative for Pediatric Drug and Device Development, San Francisco, CA 94143, USA; 6Department of Neurology, Weill Institute for Neurosciences, University of California San Francisco, San Francisco, CA 94158, USA; 7Department of Biomedical Sciences, University of California San Francisco, San Francisco, CA 94143, USA

**Keywords:** ovine model, neonates, brain hypoxia-ischemia, clemastine, asphyxia

## Abstract

Originally approved by the U.S. Food and Drug Administration (FDA) for its antihistamine properties, clemastine can also promote white matter integrity and has shown promise in the treatment of demyelinating diseases such as multiple sclerosis. Here, we conducted an in-depth analysis of the feasibility, safety, and neuroprotective efficacy of clemastine administration in near-term lambs (*n* = 25, 141–143 days) following a global ischemic insult induced via an umbilical cord occlusion (UCO) model. Lambs were randomly assigned to receive clemastine or placebo postnatally, and outcomes were assessed over a six-day period. Clemastine administration was well tolerated. While treated lambs demonstrated improvements in inflammatory scores, their neurodevelopmental outcomes were unchanged.

## 1. Introduction

Clemastine is an FDA-approved antihistamine that also exhibits anti-muscarinic activity. Via H1-receptor (H1R) antagonism, clemastine impacts mast cell function to limit allergic responses [1]. Following brain ischemia, mast cells proliferate at the injury site, and their number rapidly and persistently increases for days and weeks, contributing to inflammation [2]. Interestingly, the H1-receptor is also expressed in the hypothalamus, hippocampus, brainstem, thalamus, and cortex of the central nervous system (CNS). In addition to affecting allergy responses, clemastine thus also demonstrates neuroprotective effects. The H1R can activate neurons by increasing intracellular Ca^2+^. In this capacity, the H1R participates in the regulation of motor function, arousal, and cognition [3,4]. Muscarinic acetylcholine receptors, similar to H1R, are developmentally regulated G protein-coupled receptors (GPCRs) broadly expressed in the CNS [4,5,6]. The highest concentrations of muscarinic M1 receptors are found in cortical layers III and V/VI regions on pyramidal neurons and the hippocampus. Located postsynaptically, they are predominantly associated with excitatory synapses but are also found at cholinergic synapses [7]. M1 receptors were also identified at the level of brain arterioles, capillaries, and venules, thus playing an important role in regulating brain perfusion [8]. In addition, M1 receptors are also expressed on microglia and may regulate microglia-OPC crosstalk during developmental myelination [9]. Clemastine is often produced in salt form with fumaric acid in order to improve its solubility and bioavailability [3]. The fumarate component of clemastine also has immune-modulatory properties altering immune cell composition and phenotype, thereby restricting CNS infiltration [10]. In this capacity, fumarate has been used with some success in the clinical treatment of multiple sclerosis [11]. Fumarate acts via nuclear factor erythroid-derived 2-related factor (Nrf2)-dependent and independent pathways that have been observed to be neuroprotective in neonatal hypoxia-ischemia induced brain injury (HI) models [12,13].

Clemastine has emerged as a promising therapeutic agent for demyelinating diseases in adults. Developmentally, myelination represents the last steps of brain formation that begin following synaptogenesis shortly before a term neonate is born [14]. Perinatal asphyxia that results in neonatal HI can thus lead to significant myelin damage in the immediate perinatal period, as well as postnatally, and result in long-term impairment in motor function [14]. The failure in myelin regeneration after HI seems to be due to both early premyelinating oligodendrocyte degeneration and a later phase of persistent premyelinating oligodendrocyte maturation arrest [15]. Clemastine has shown greatest therapeutic potential in demyelinating brain conditions, such as multiple sclerosis [16], hypoxia-induced white matter injury [17], and spinal cord injury [18]. Clemastine can also decrease microgliosis and enhance motor neuron survival in amyotrophic lateral sclerosis, while alleviating neuroinflammation and improving cognition in Alzheimer’s disease [3]. In rodent studies of neonatal HI, clemastine significantly reduced brain area loss following daily post-hypoxic treatments [19]. In vitro, clemastine significantly promoted differentiation of purified OPCs cultured in isolation as well as myelination when OPCs were co-cultured with purified dorsal root ganglion neurons [15]. The remyelinating properties stem from clemastine’s ability to promote differentiation of oligodendrocyte progenitor cells through the activation of ERK1/2 via muscarinic receptors [18]. Clemastine also prevents OPCs from entering cellular senescence [20]. These effects of clemastine on OPCs lead to preservation of myelin integrity, decrease in axonal loss, and delays in axonal degeneration [21]. Most importantly, success of clemastine as a remyelination therapy has been noted in clinical trials of relapsing multiple sclerosis with chronic demyelinating optic neuropathy [22]. 

Clemastine thus holds promise as a potential neuroprotective therapy for HI in term infants as well as white matter injury in preterm neonates. Its anti-inflammatory effects, enhancement of neuroplasticity and synaptogenesis, modulation of neurotransmitter systems, stabilization of the BBB, and promotion of myelination and remyelination are potential mechanisms by which clemastine could exert its neuroprotective effects in this patient population. This study focused on defining the pharmacokinetics, safety, and efficacy of clemastine administration in a large animal model of neonatal HI.

## 2. Materials and Methods

### 2.1. Animals

The data in this work is available upon request. All animal research was approved by the University of California Davis Institutional Animal Care and Use Committee and was performed in accordance with the Guide for the Care and Use of Laboratory Animals. The sheep were kept in conditions regulated by the Guide for the Care and Use of Agricultural Animals Used in Research (Ag Guide), USDA Animal Welfare Regulations (AWR), and the Guide for the Care and Use of Laboratory Animals (Guide). All handling for research was approved by the Attending Veterinarian and the IACUC. White Dorper sheep were purchased from Luke Vanlaningham at ~133 days gestation. The ewes were maintained in a barn where they were fed alfalfa and grass hay twice daily and had access to fresh water. The ewes were not fed the night before the UCO. After the UCO the ewes were euthanized. The lambs were kept on a radiant warmer to maintain normothermia until weaned from mechanical ventilation and extubation. After extubation, the lambs were placed in a pen to recover. Lambs were kept under carefully regulated conditions to ensure their well-being and health and to minimize potential sources of variability and stress that could impact the research findings. The housing and care of these animals strictly adhered to ethical and regulatory standards. The lambs were housed in a controlled environment where factors such as temperature, humidity, and lighting were closely monitored to provide a comfortable and stress-free setting. The lambs were subject to Q4-6h observations by trained personnel to check for any signs of health concerns or distress, allowing for early detection of any health concerns and for assessment of neurodevelopmental outcomes. Feeding schedule followed tube feeding within 3 h following resuscitation and thereafter until lambs were able to bottle feed. The lambs were given a commercially available colostrum replacer the first 24 h and then transitioned to a commercially available milk replacer. They were housed in a 5 ft by 5 ft pen in the same barn the ewes had been maintained in on rubber mats and shavings and had two heat lamps turned on during the 10 p.m. feeding and turned off again at 8 a.m. to facilitate the milk digestion. For the scope of this study, we did not use hypothermia. Study participants as well as laboratory and data analysis personnel were blinded to treatment group throughout the duration of the study.

### 2.2. Umbilical Cord Occlusion Model

The study used white Dorper sheep of both sexes. HI was induced via UCO in near term lambs at 141–143 days gestation (term ~147–150 days). Pregnant ewes, whose gestational timing was precisely determined, underwent a fasting period of 12–24 h before the surgical procedures. Anesthetic induction of the ewes was performed using ketamine and propofol and anesthesia level was maintained with 1–5% isoflurane through the endotracheal tube according to IACUC approved Standard Operating Procedure, SC-20-112, “Sheep Anesthesia: Surgical Research Facility, H-Building at TRACS”. After anesthetic induction, a jugular catheter or peripheral venous catheter was placed, the ewe given 4 mg/kg slow push IV propofol, and 1–5 mg/ketamine, followed by intubation. Immediately prior to surgery, the pregnant ewes underwent ultrasound imaging to confirm pregnancy status. Following the imaging, the ventral abdomen was scrubbed using either Betadine or Chlorhexidine and alcohol. The ewe was started on maintenance intravenous (IV) fluids at the rate of 5–15 mL/kg/h. The oxygen levels in the ewe were continuously monitored using an O_2_ saturation probe, while hemodynamics was closely observed using a noninvasive blood pressure cuff. A midline incision, measuring approximately 6–10 inches, was surgically created along the ventral abdomen, allowing access to the uterus. To prevent infection, the ewe received intravenous antibiotics in the form of penicillin G potassium at a dosage of 10,000–20,000 units/kg and gentamicin at a dosage of 1–2 mg/kg. Following the externalization of the fetal head, the fetus was intubated using a properly sized cuffed endotracheal tube (ETT). Gravity assisted in the passive drainage of lung liquid, while the ETT was plugged to prevent gas exchange during gasping episodes. Additionally, venous and arterial catheters were inserted into the jugular vein and carotid artery to facilitate hemodynamic monitoring, blood sampling, and the administration of medications. Asphyxia was induced by UCO until the onset of asystole. The umbilical cord was cut, and the lamb was transitioned to a radiant warmer. After a 5 min period of asystole, confirmed through invasive hemodynamic monitoring, resuscitation of the lambs was initiated with positive pressure ventilation with a fraction of inspired oxygen FiO_2_ of 1.0. Resuscitation was not initiated with room air as asystole requiring chest compressions is universal given the severity of the model; therefore, oxygen therapy was clinically indicated. Following 30 s of ventilation, external chest compressions were initiated, lasting for 60 s intervals. The heart rate was reassessed at the end of each interval, and these resuscitative measures were sustained for a maximum duration of 15 min. Intravenous administration of epinephrine at a dose of 0.01 mg/kg was given when an insufficient response was observed despite oxygen, ventilation, and chest compressions after 60 s of asystole post-initiation of ventilation. Additional doses of epinephrine were administered if the initial doses failed to yield a response. Volume boluses were not administered, as the combination of oxygen, epinephrine, and chest compressions typically proved effective in restoring adequate perfusion following the ROSC (sustained heart rate of 100 bpm (beats per minute) and SBP > 20 mmHg). Throughout and following the resuscitation, a mechanical ventilator was utilized to provide continuous mechanical ventilation. Weaning from assisted ventilation was initiated, and it was eventually discontinued when the lamb exhibited spontaneous breathing for more than 50% of the time and maintained a peripheral oxygen saturation level exceeding 85% while receiving an FiO_2_ of 0.21. No intravenous fluids were administered either during or after the resuscitation. Following extubation, the lambs were initially fed 2 oz through a tube every 4 h on the first day. Subsequently, the feeding regimen consisted of 2–6 oz via bottle, with the quantity adjusted based on the lamb’s size, provided four times a day. In cases where the lamb was unable to bottle feed, tube feeding was continued at a rate of 2–4 oz four times a day, alongside daily attempts at bottle feeding. Neurodevelopmental outcomes of the lambs were evaluated over a 6-day period, followed by euthanasia on day 6 with an overdose of 100 mg/kg pentobarbitone sodium solution (Lethabarb^TM^, Virbac Pty. Ltd., Peakhurst, NSW, Australia).

### 2.3. Drug Treatment and Pharmacokinetic Analysis

Randomization was performed at the level of the lamb. Animals in the placebo arm received an equal volume of normal saline IV infusion at matched time points. Study drug was prepared by an individual separate from the research team to allow the research team to remain blinded to treatment assignment. Dose escalation study: Two lambs received escalating doses of 0.6 mg/kg, 2 mg/kg and 6 mg/kg IV clemastine over 10 min starting 10 min after delivery and at 24 and 48 h of life. The lambs did not undergo umbilical cord occlusion. Plasma samples were collected for pharmacokinetic analysis from the lambs prior to each infusion, at the end of each infusion, and at 1, 2, 4, 8, 24, and 48 h following infusion. Clemastine in lamb plasma was quantified using LC-MS with a lower limit of quantification (LLOQ) of 0.01 ng/mL. Values below the LLOQ were imputed at half the LLOQ for analysis. Noncompartmental analysis (NCA) was performed using R package ‘NonCompart’ v0.6.0 in R v4.2.0 (Bae 2022).

### 2.4. Neurobehavioral Outcomes

We evaluated the number of days required for the attainment of typical lamb behavioral milestones following birth (head lift and shake; use of front and hind limbs; use of four legs; standing; walking) for a cumulative total score of 4 (Table 1). Ability to feed and activity at rest were evaluated separately and were reported as a sum score of 2. The severity of impairment was assessed based on the composite score of motor function, feeding, and activity.

The severity of impairment was classified based on a composite score of motor function, feeding, and activity. The highest score represents no impairment of the selected neurological function.

### 2.5. Biochemical Markers of Inflammation

We collected complete blood count prior to the UCO (BSN) at 8 h, and on days 1, 2, 3, 5, and 6. We assessed the differences in white blood cells (WBC), neutrophils (absolute neutrophil count, ANC), lymphocytes (absolute lymphocyte count, ALC), platelets (PLT), monocytes (Mono), and eosinophils (Eos). We calculated system inflammation response index (SIRI) = ANC × (Mono/ALC), systemic immune inflammation index (SII) = PLT × (ANC/ALC).

### 2.6. Power Analysis, Sample Size Calculation, Attrition Rate, Blinding, and Randomization

The sample size was set at 10 lambs per treatment arm, corresponding to 80% power at detecting an 80% relative risk reduction (RRR) in a test of two, ewe-clustered (intracluster correlation = 0.1) binomial proportions with 65% control event and a 5% type-I error rate (one-sided), assuming each ewe gave birth to either one or two lambs with equal probabilities. These numbers already accounted for the type-II error rate inflation resulting from futility testing after the first seven outcomes from each group had become available and assumed 15% attrition rate. Early stopping for futility would happen if the *p*-value at the interim analysis exceeded the value of 0.2822. Researchers performing experiments and analyzing data were blinded to groups. Randomization using an envelope system was applied. Envelopes were prepared for each lamb in advance, each containing a slip of paper indicating which treatment group the lamb would be assigned to. The study is a double-blind, as the researchers performing experiments and animal care, as well as the data analysts, were all unaware of the treatment group assignments. The researchers performing all the experiments, including the injury, post-injury care, biochemical, histological, and neurological outcomes analysis were blinded to the group assignment until all measurements had been collected. The data was separated into two groups, data analysis was performed, and the identity of the groups was revealed after the analysis was performed.

### 2.7. Statistical Analysis

Analyses of data were performed using Prism 10 (version 10.0.3), GraphPad Software, San Diego, CA, USA). All data are shown as mean  ±  standard error of measurement. Differences were considered significant at *p*  <  0.05. Data was subjected to a normality test. If the data passed the normality test, the differences between two groups were assessed by t-tests; otherwise, we applied the Mann–Whitney test. Grouped data were analyzed using one-way and two-way analysis of variance and subsequently subjected to correction for multiple comparison analyses. The hemodynamic data were analyzed using grouped analysis of the individual group’s means for a specific time point. Comparisons were made between clemastine-treated (Clemastine) and untreated placebo groups (Placebo), male and female sex.

## 3. Results

### 3.1. Clemastine Pharmacokinetics and Dose Selection

Allometric scaling of the adult human dose of 4–5.5 mg clemastine PO twice daily used for neuroprotection in multiple sclerosis and other neurologic conditions, and assuming 40% oral bioavailability, resulted in an equivalent dose of 0.1–0.2 mg/kg/day IV for a 3 kg lamb [15,23]. Predicted Cmax and AUC for an adult male following a single 4 mg/kg PO clemastine dose were 1.7 ng/mL and 48 ng*h/mL, respectively [23]. However, most rodent studies showing neuroprotection have used substantially higher doses of clemastine. Allometric scaling of 1 mg/kg IP injection used in rodent stroke models (250 g adult rat) which showed neuroprotection, and including a 25% increase to account for IV instead of IP dosing, resulted in an equivalent dose of 0.7 mg/kg IV for a 3 kg lamb [24]. Allometric scaling of the 10 mg/kg PO dosing used in other rodent models including periventricular leukomalacia in 15 g mouse pups and assuming 40% oral bioavailability resulted in an equivalent dose of 1.0 mg/kg IV for a 3 kg lamb [15,23]. Some studies have used doses of clemastine as high as 10 mg/kg IP, which scales to 6 mg/kg in a 3 kg neonatal lamb after accounting for differences between IV and IP administration. We therefore conducted a dose escalation study evaluating the pharmacokinetic profile of 0.6 mg/kg, 2 mg/kg and 6 mg/kg clemastine IV in our model. Clemastine concentration was measured in 30 samples from two lambs. Clemastine concentrations following the 6 mg/kg dose were only available from one lamb due to technical issues with sample collection. Maximal plasma clemastine concentration occurred at the end of each infusion and ranged from 0.185 mg/L following a 0.6 mg/kg dose and 27.5 mg/L following a 6 mg/kg dose (Figure 1A). AUC ranged from 0.63 to 20.881 mg*h/L, with higher exposure seen after higher doses and 0.253–0.852 h/L when normalized by dose. Median elimination half-life was 5.3 h.

In terms of toxicity, in the dose escalation study, both lambs were observed to be sedated with decreased spontaneous activity for approximately 15 min following the 2 mg/kg dose and >1 h following the 6 mg/kg dose. No sedation or change in spontaneous activity was observed following the 0.6 mg/kg dose. Mild tachycardia was observed following the 2 mg/kg dose (HR 120–150) in 2/2 lambs. No seizures, dry mucous membranes, urinary retention, hyperthermia, or mydriasis were observed following any dose in the dose escalation study. A dose of 2 mg/kg was chosen as the study dose based on these results.

Following the dose escalation study, toxicity data was reviewed for twelve additional lambs that underwent umbilical cord occlusion and were randomized to receive 2 mg/kg IV clemastine (n = 5) or placebo (n = 7) infused over 10 min starting 10 min following resuscitation and repeated q24h x3 total doses. Clemastine pharmacokinetic parameters are likely similar in neonatal lambs and young children and plasma levels achieved following 2 mg/kg IV clemastine infusion in lambs were substantially higher than those that have been shown to be neuroprotective in adults with MS. However, a therapeutic range for neuroprotection in human neonates has not been defined.

### 3.2. Clemastine Safety and Toxicity

Some adverse events, primarily sedation, were seen following clemastine infusion at higher doses. No dry mucous membranes, urinary retention, hyperthermia, or mydriasis were observed. Evaluating resuscitation outcomes, we have not noticed significant sex differences in the hemodynamic response to the UCO. The mean arterial pressure (MAP) was slightly elevated in the clemastine-treated females compared to placebo females at 36–40 min after the UCO (Figure 1B). Clemastine treated animals had higher systolic blood pressure (SBP) and mean arterial pressure (MAP) compared to placebo (Figure 1B); however, this did not impact the onset of return of spontaneous circulation (ROSC) (Figure 1C) and was observed prior to clemastine administration. We did not observe differences in time from UCO to asystole (Figure 1C). The placebo and clemastine treatment groups did not significantly differ in their requirement for epinephrine administration during resuscitation (Figure 1D). In total, 0/10 lambs in the clemastine group and 1/15 lambs in the placebo group exhibited clinical seizure activity following resuscitation (Figure 1D). The requirement for dextrose infusion to treat hypoglycemia after resuscitation was similar between the studied groups (n = 5/10 in the clemastine group, n = 9/15 in the placebo group) (Figure 1D). Mortality between the studied groups was not different (n = 1 in placebo group) (Figure 1E). The sex representation between the groups was similar (Figure 1E). The sex did not have an effect on anthropometric measurements in our study (Figure 1F). The studied groups had similar body weight at birth (Figure 1F). Clemastine administration did not affect body and brain weight at explant (Figure 1F).

### 3.3. Clemastine Effect on Biochemical Markers

Serial arterial blood gasses were drawn at baseline pre-UCO (BSN), immediately before initiation of CPR following delivery, and at 10, 20, 30, and 60 min after ROSC. Consistent with our prior studies [25], the UCO protocol produced a clinically significant combined metabolic and respiratory acidosis in all lambs (Figure 2A). Sex had no significant impact on any of the parameters studied (Figure 2A,B). There were no significant changes in oxygenation and ventilation among the studied groups (Figure 2A). Compared to vehicle-treated lambs, the clemastine-treated lambs had slightly higher glucose levels at baseline (*p* = 0.04). Placebo animals experienced higher glucose levels at 10, 20, and 30 min after the resuscitation (*p* = 0.02, *p* = 0.04 and *p* = 0.01) (Figure 2A). Evaluation of end-organ injury biochemical markers revealed elevated BUN in the clemastine group on day 1 following UCO (*p* = 0.005), whereas creatinine at baseline and 1 h after UCO were slightly lower (*p* = 0.01 and *p* = 0.02), with no significant differences noted thereafter.

### 3.4. Clemastine Effect on Peripheral Blood Cells and Inflammatory Indices

We assessed for differences in white blood cells (WBC), neutrophils (absolute neutrophil count, ANC), lymphocytes (absolute lymphocyte count, ALC), platelets (PLT), monocytes (Mono), and eosinophils (Eos). We calculated systemic inflammation response index (SIRI) = ANC × (Mono/ALC), systemic immune inflammation index (SII) = PLT × (ANC/ALC).

We observed higher WBC counts on day 1 after UCO in placebo males compared to placebo females (*p* = 0.01) and clemastine-treated males (*p* = 0.04). Compared to placebo females, placebo males also exhibited elevated ANC (*p* = 0.03) and SII scores (*p* = 0.03) on day 1 after the UCO. At baseline (pre-UCO), clemastine-treated females exhibited higher ALC levels compared to placebo females (*p* = 0.03). The clemastine-treated group exhibited slightly higher ALC at baseline compared to placebo lambs (*p* = 0.01). Clemastine-treated lambs demonstrated lower ANC on day 5 after UCO (*p* = 0.006). We did not observe changes in other subgroups of peripheral blood cells (Figure 3). Both peripheral blood cell indices, SII and SIRI, were decreased in clemastine-treated lambs on day 5 (Figure 3) (SII, *p* = 0.01; SIRI, *p* = 0.007).

### 3.5. Clemastine Effect on Neurological Outcomes

We evaluated whether clemastine treatment produced clinically relevant improvements in neurodevelopmental outcomes in lambs subjected to HI. Specifically, we assessed motor function, activity at rest, and ability to feed. We did not detect an impact of sex on neurodevelopmental outcomes (Figure 4A). All injured animals demonstrated significant encephalopathy in the first 24 h after the UCO. While clemastine treatment demonstrated a trend towards improvement in neurodevelopmental outcomes on day 1 after UCO, the overall outcomes were not different between the studied groups. Clemastine treated lambs demonstrated a trend toward worse feeding skills after day 1, as well as motor milestones on day 6 after UCO (Figure 4A).

### 3.6. Markers Associated with Total Neurological Outcomes Score

We also assessed whether any of the parameters investigated in this study were associated with poor neurological outcomes on day 6 following UCO. We first evaluated whether there are sex-specific associations with total neurological outcomes on day 6. In males, higher ALT on day 1 correlated with poor outcomes on day 6 (*p* = 0.003). Poor outcomes in the subscores of individual neurological outcomes, such as motor function and activity, correlated with poor outcomes on day 6 from day 2 after UCO. Feeding score did not show a correlation with total outcomes on day 6. In females, time from UCO to ROSC was the earliest predictor of poor outcomes on day 6 (*p* = 0.001). From biochemical markers, higher BUN and BUN/Cr ratio on day 3 (*p* = 0.03 and *p* = 0.04), AST on day 6 (*p* = 0.03), and platelets on day 6 (*p* = 0.03) correlated with worse outcomes on day 6. Worse total outcomes, activity, and motor outcomes already observed on day 1 after UCO correlated with worse total outcomes on day 6 (0.007, 0.001 and 0.02). Assessing both sexes together, the strongest correlation was identified in the time from UCO to ROSC (*p* = 0.005). Total outcomes evaluated on day 1 were also correlated with total outcomes on day 6 (*p* = 0.0047). Similarly, motor outcomes and activity after day 1 (*p* = 0.0001 and *p* < 0.0001) were correlated to total outcomes on day 6. Feeding skills correlated with total outcomes only on day 6 (*p* = 0.04). From the biochemical markers, BUN levels showed correlation on day 3 (*p* = 0.04). BUN on day 1 (*p* = 0.07), monocyte levels on day 6 (*p* = 0.06), and glucose levels at 60 min (*p* = 0.06) were additional markers closest to significance (Figure 4B).

## 4. Discussion

Clemastine is considered safe in children when used for other indications at doses up to 0.5 mg PO twice daily. While effective for demyelinating disorders in adults, the optimal dose of clemastine for neuroprotection in neonates is unknown. Clemastine can be administered by injection, orally, or topically. Clemastine binds to plasma proteins (70–97%) and is then metabolized by the liver and mostly excreted in the urine within 24 h of intake [26]. The elimination half-life can be variable, especially in sick neonates after an asphyxia event. As a result of clemastine’s high binding to plasma proteins, and due to its active metabolites such as fumarate, the duration of the pharmacological effect of clemastine could be much longer than the plasma half-life [26]. As a first-generation antihistaminic, clemastine has poor receptor selectivity and non-specifically binds muscarinic, serotonin, and α-adrenergic receptors, as well as cardiac potassium ion channels, that are responsible for potential adverse effects [27]. Due to high lipid solubility, clemastine can easily cross the blood–brain barrier (BBB) and induce sedation [26]. In our previous studies, animals that were injured consistently exhibited some degree of encephalopathy following UCO. Interestingly, a subgroup of animals showed improvements in encephalopathy by day 2–3, while another subgroup exhibited persistent encephalopathy, particularly in cases of more severe injuries [25]. In this study, clemastine treatment did not significantly impair the alertness of animals compared to the placebo group, except soon after infusion. Increased drowsiness associated with refusal to eat has been reported in infants following clemastine administration to breast-feeding mothers [28]. Although there was no significant difference in the feeding ability between the groups in our study, we did observe a trend towards poorer feeding skills in the clemastine-treated groups. It is important to note that this observation warrants thorough evaluation in a clinical setting, as neonatal HI itself can lead to feeding difficulties that may result in aspiration events [29]. We have further evaluated hemodynamic effects of clemastine administration. Clemastine has been demonstrated to suppress hERG K^+^ channels [30] which could potentially result in QT-interval prolongation and associated dysrhythmias. Cardiovascular dysfunction, including depressed biventricular function, as well as pulmonary arterial and venous hypertension as described in neonates following a perinatal hypoxic insult [31], can create a substrate for arrhythmias. Although able to demonstrate torsadogenic potential in a rabbit AV block model [32], we did not observe a higher incidence of arrhythmias in the immediate post resuscitative period following clemastine infusion compared with placebo, consistent with its longstanding safety profile. 

UCO led to similar changes in blood gas parameters, such as severe metabolic acidosis, reflected by low pH, high lactate levels, and profound base deficit that are seen in newborn babies after birth asphyxia. In lambs treated with clemastine, baseline glucose levels were consistently lower, both before and at all timepoints studied after UCO. It is worth noting that first-generation antihistamines have been reported to affect glucose levels [33]. Administration of clemastine resulted in a modest decrease in fasting glucose levels and an increase in post-prandial blood glucose levels [33]. In fetal lambs, glucose levels rise proportionally with the release of cortisol and catecholamines [34], and the higher glucose levels observed in placebo-treated lambs might indicate an intensified stress response following UCO. We also noticed isolated elevation of the BUN on day 1 after UCO in the clemastine-treated group and elevated creatinine in the placebo group at baseline and 1 h after UCO. It’s important to note that H1 receptors are expressed on renal vessels [35], which leads us to speculate that the administration of clemastine may possess some renal protective properties in the immediate period after the UCO.

UCO induces ischemic-reperfusion injury, which initiates systemic as well as localized immune responses within the brain parenchyma. These responses evolve over time and exhibit developmental specificity. Immune dysregulation during the first week after stroke includes changes in peripheral blood cell ratios and cell immunotypes that can be predictive of severity of neurological outcomes [36,37,38,39]. Lymphocytes localize to the brain in the acute phase of cardiac arrest resuscitation and may directly damage neurons and associated cells or stimulate a harmful pro-inflammatory microenvironment [40]. Clemastine had no effect on lymphocyte counts, but it did influence neutrophil counts at later time intervals. Unlike in adult stroke cases, where the initial immune response is driven by neutrophils infiltrating the injury site, neonates primarily rely on activated resident microglial cells as the key immune cells instigating the pro-inflammatory cascade [41]. Neutrophils, however, represent the majority of infiltrating cells after stroke in neonates, and increased neutrophil infiltration was associated with increased hypoxic-ischemic brain injury [42,43]. A noteworthy finding in our study was lower neutrophil counts in clemastine-treated animals that could contribute to neuroprotection. Clemastine-treated animals also demonstrated lower SII and SIRI scores. The SII and SIRI associate with stroke severity and SII was suggested to be a superior predictor of unfavorable functional outcomes [44]. SII reflects the pro-thrombotic state due to elevated platelet counts, immune dysregulation and inflammation with higher neutrophil and lower lymphocyte counts triggered by stroke [45]. We identified attenuated SIRI and SII indices in clemastine-treated animals on day 5 after UCO. These findings suggest that clemastine treatment may contribute to modifying the immune cell response and inflammation, potentially mitigating HI injury.

Our observations revealed that there were no discernible enhancements in neurodevelopmental outcomes between the group of animals treated with clemastine and the placebo group. Furthermore, we did not identify any variations in outcomes based on the animals’ sex. The levels of encephalopathy and vigorousness were comparable between the studied groups. While reduced alertness might potentially impact feeding abilities, we did not detect any significant differences in feeding between the groups. Likewise, there were no noteworthy distinctions in motor outcomes between the treatment groups. Although our study did not reveal substantial enhancements in neurological outcomes in near-term lambs following the UCO, it highlights the necessity to explore clemastine’s broader therapeutic potential. Clemastine exhibits a diverse range of desirable mechanisms of action that have demonstrated neuroprotective effects both in vitro and in vivo. Clemastine has immunomodulatory properties on a systemic level [46], as well as local level by regulation of microglial responses [47], attenuation of neuronal apoptosis [47], and astrocytes loss [48]. One of the most significant effects of clemastine is increase in oligodendrocyte differentiation and enhancement of remyelination shown in various animal models [48]. Most importantly, success of clemastine as a remyelination therapy has been noted in clinical trials of relapsing multiple sclerosis with chronic demyelinating optic neuropathy [22]. In HI, oligodendrocytes are particularly vulnerable to hypoxia and hypoglycemia [49]. Evidence suggests that in attempts to rescue demyelinated areas following a stroke, there is an increase in the density of perivascular oligodendrocyte progenitor cells [49]. In term infants in particular, myelinated fibers are more metabolically active and hence more vulnerable to HI [50]. This vulnerability makes clemastine an ideal neurotherapeutic candidate, considering clemastine’s demonstrated potential to protect and restore myelin, along with its favorable pharmacokinetic profile and minimal toxicity confirmed by our study. Thus, further investigation is warranted, ideally in a clinical setting, to explore clemastine as a neuroprotective agent for HI.

Finally, we conducted an evaluation to determine if any of the examined parameters exhibited correlations with overall neurological outcomes on day 6. The most notable early correlation we identified was between the interval from UCO to ROSC. It is evident that the duration of the insult had a substantial impact on subsequent neurological outcomes. Additionally, we observed that low total scores on day 1, as well as diminished motor skills and activity after day 2 following UCO, were associated with unfavorable neurological outcomes. We also observed correlation between BUN/Cr ratio on day 6 and poor total neurological outcomes. This finding aligns with observations in human ischemic stroke, where an elevated BUN/Cr ratio serves as a prognostic indicator for less favorable neurological outcomes [51,52].

There are several potential explanations for the observed limited neuroprotection with clemastine. One reason could be the relatively short follow-up duration. While a 6-day follow-up period is longer than what’s typical in many large animal studies, it may still only capture a transitional phase between tissue injury and the initiation of repair mechanisms. Therefore, longer observation periods might be necessary to fully understand and appreciate the impact of clemastine on neurological outcomes. Additionally, the relatively small sample size in our study limited our ability to assess potential gender differences, which are known to be an established risk factor for adverse neurological outcomes [53,54].

## 5. Conclusions

The evaluation of clemastine as a potential neuroprotective agent in neonates at risk of HI has revealed several findings and considerations. While clemastine has shown therapeutic promise for demyelinating disorders and in several animal models of HI, its use in neonates for neuroprotection remains an area of active investigation. Our study in neonatal lambs mimics the pharmacokinetics in young children following 2 mg/kg IV clemastine infusion, a dose substantially higher than those that exhibited neuroprotection in adults with MS. Clemastine’s wide receptor selectivity could lead to potential adverse effects, most importantly impaired alertness and arrhythmias. Despite UCO itself being a trigger for altered mental status and impaired hemodynamics due to severe hypoxia, hypercarbia, and acidosis, administration of clemastine did not lead to concerning side effects or end-organ toxicity in our model. Clemastine may exhibit its effects by modulation of immune responses, and these effects need to be understood and defined in in-human clinical trials after HI. While clemastine administration did not translate into significant improvements in neurodevelopmental outcomes in our study, the improvement of select biochemical and inflammatory markers warrants comprehensive assessment of clemastine’s potential as neurotherapeutic for HI by further studies with longer follow-up durations, larger sample sizes, and consideration of potential gender differences.

## Figures and Tables

**Figure 1 children-10-01728-f001:**
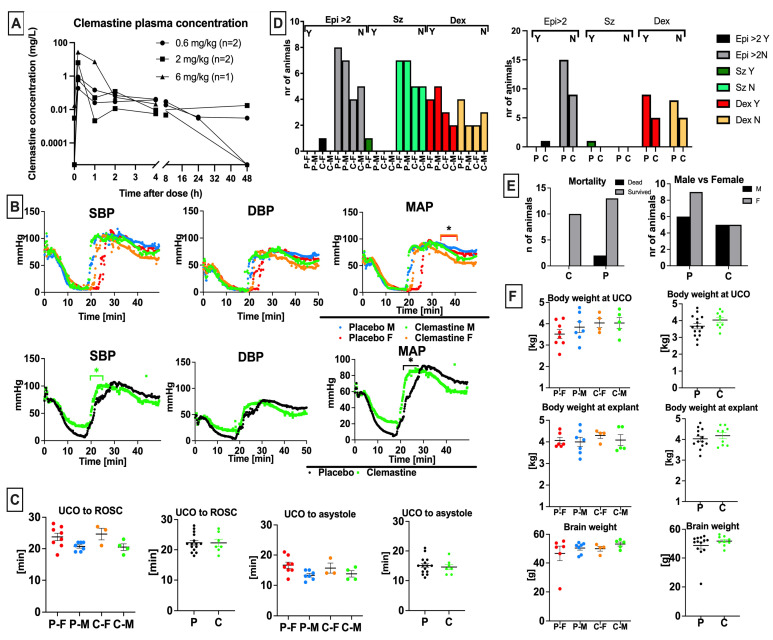
Clemastine pharmacokinetics and resuscitation outcomes: (**A**) Concentration-time profiles of clemastine levels in lamb plasma. (**B**) Clemastine females achieved higher MAP compared to placebo females in the immediate post-resuscitative period. Clemastine-treated animals also achieved higher SBP and MAP briefly after the UCO, (**C**) ROSC, and time from UCO occlusion to asystole; as well as (**D**) the incidence of 2nd dose of epinephrine and dextrose administration were similar between the studied groups. (**E**) Mortality and proportion of male and female sex were similar between the studied groups. (**F**) There were no differences in selected anthropometric parameters between clemastine and placebo groups. CPR—cardiopulmonary resuscitation; UCO—umbilical cord occlusion; ROSC—return of spontaneous circulation; SBP—systolic blood pressure; MAP—mean arterial pressure; DBP—diastolic blood pressure; C—clemastine-treated group, green. P—placebo, black. Controls—yellow; P-F—placebo females, red; P-M—placebo males, blue; C-F—clemastine females, orange; C-M—clemastine males, green. * *p* < 0.05.

**Figure 2 children-10-01728-f002:**
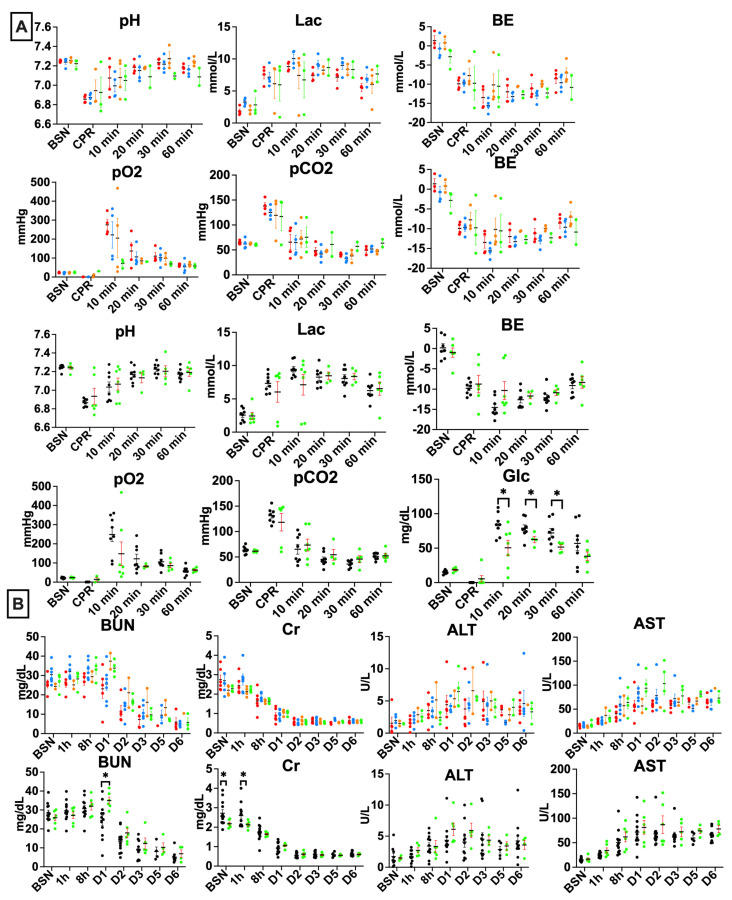
Biochemical changes in response to the UCO: (**A**) UCO leads to profound acidosis in both clemastine-treated group, as well as placebo, compared to controls. The clemastine and placebo groups did not differ in levels of hyperlactatemia, base deficit, oxygen, or carbon dioxide levels. Clemastine-treated lambs experienced higher glucose levels at the baseline, whereas placebo lambs had persistent hyperglycemia as a result stress response to the UCO. (**B**) End-organ function markers showed elevated BUN in the clemastine group on D1 after the UCO. Creatinine was elevated in the placebo group at the baseline and 1 h after the UCO. Data in the graphs are mean ± SEM. For (**A**): Clemastine: n = 7–8, Placebo: n = 7–8; for (**B**): Clemastine:6, Placebo:12. preUCO-baseline pre-UCO; CPR—cardiopulmonary resuscitation; glc—glucose; BE—base excess; lac—lactate; BUN—blood urea nitrogen; Cr—creatinine; AST—aspartate aminotransferase; ALT—alanine transaminase. Clemastine-treated group, green, placebo, black. P-F—placebo females, red; P-M—placebo males, blue; C-F—clemastine females, orange; C-M—clemastine males, green. * *p* < 0.05.

**Figure 3 children-10-01728-f003:**
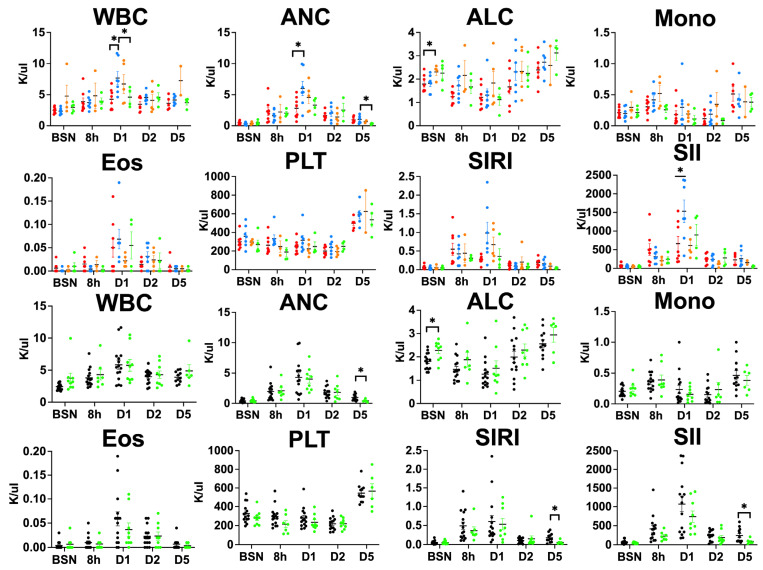
Peripheral markers of inflammation: In the clemastine-treated groups, ALC differed at baseline and ANC on day 5 after the UCO. The inflammatory indices, SIRI and SII, were decreased in the clemastine-treated lambs compared to the placebo group on day 5. The summary column graphs are showing means ± SEM. WBC—white blood cells; ALC—absolute lymphocyte count; ANC—absolute neutrophil count; Mono—monocyte; Eos—eosinophils; PLT—platelets; SIRI—system inflammation response index; SII—systemic immune inflammation index. P-F—placebo females, red; P-M—placebo males, blue; C-F—clemastine females, orange; C-M—clemastine males, green. Clemastine-treated group, green. Placebo, black. * *p* < 0.05.

**Figure 4 children-10-01728-f004:**
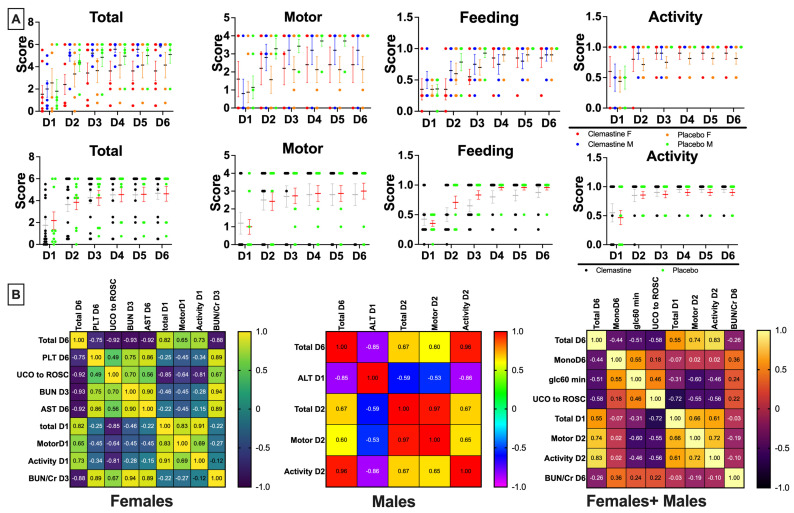
Neurological outcomes: (**A**), we assessed total outcomes score, as a composite score consisting of motor function and feeds + activity. Clemastine- treated group, green. Placebo- black. (**B**), The correlation matrix pictures Spearman’s correlation coefficients of selected study parameters with combined neurological outcomes scores on day 6 after the UCO that reached statistical significance (*p* < 0.05). We report also markers that did not reach statistical significance, BUN day1 (*p* < 0.07), glc at 60 min (*p* = 0.06) and Mono on day 6 (*p* = 0.06). UCO—umbilical cord occlusion, WBC—white blood cells, ROSC—return of spontaneous circulation, BUN—blood urea nitrogen, glc—glucose, Mono—monocytes.

**Table 1 children-10-01728-t001:** Neurobehavioral assessment score.

Function	Neurological Milestone	Score
Motor function	Walking	4
	Standing	3
	Four limbs	2
	Front/hind limbs	1
	No movement/Spastic	0
Feeding	Nurses normally	1
	Suckling well once finds the bottle	0.5
	Requires assistance to find bottle; a few good suckles	0.25
	Minimal suckle, tube fed	0
Activity at rest	Lifts the head up, alert active	1
	Wakes up with stimulation, attempts to lift the head	0.5
	Sleepy; no head lift with stimulation	0

## Data Availability

Data are contained within the article. Further enquires can be directed at the corresponding author.

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
