# Peer review of "Effect of Clemastine on Neurophysiological Outcomes in an Ovine Model of Neonatal Hypoxic-Ischemic Encephalopathy"

_children, 2023, doi:10.3390/children10111728_

Round 1

Reviewer 1 Report

The study investigated the feasibility, safety, and neuroprotective efficacy of clemastine administration in near-term lambs using a global ischemic insult model induced via umbilical cord occlusion. The study employed a randomized design with lambs being assigned to receive either clemastine or placebo postnatally. Outcomes were assessed over a six-day period.

The text implies that clemastine administration might have potential therapeutic benefits for demyelinating diseases, although it did not improve neurological outcomes in this particular study. The significance of the findings should be discussed in the context of the current literature on demyelinating diseases, potential mechanisms of action of clemastine, and the need for further research.

Overall, while the study provides valuable insights into the feasibility, safety, and potential benefits of clemastine in a unique animal model, several areas require clarification and additional information. Addressing these points would strengthen the scientific rigor and impact of the study.

Please add the chapter of conclusions.

Author Response

Reviewer 1:

The study investigated the feasibility, safety, and neuroprotective efficacy of clemastine administration in near-term lambs using a global ischemic insult model induced via umbilical cord occlusion. The study employed a randomized design with lambs being assigned to receive either clemastine or placebo postnatally. Outcomes were assessed over a six-day period. The text implies that clemastine administration might have potential therapeutic benefits for demyelinating diseases, although it did not improve neurological outcomes in this particular study.

The significance of the findings should be discussed in the context of the current literature on demyelinating diseases, potential mechanisms of action of clemastine, and the need for further research. Overall, while the study provides valuable insights into the feasibility, safety, and potential benefits of clemastine in a unique animal model, several areas require clarification and additional information. Addressing these points would strengthen the scientific rigor and impact of the study.

Thank you for this comment. We have edited the discussion to specifically discuss the findings in the context of clemastine’s mechanisms of action, it’s effects on myelination, and the need for further research in the context of newborn brain injury:

-p.13 Ln 450

“Our observations revealed that there were no discernible enhancements in neurodevelopmental outcomes between the group of animals treated with clemastine and the placebo group. Furthermore, we did not identify any variations in outcomes based on the animals' sex. The levels of encephalopathy and vigorousness were comparable between the studied groups. While reduced alertness might potentially impact feeding abilities, we did not detect any significant differences in feeding between the groups. Likewise, there were no noteworthy distinctions in motor outcomes between the treatment groups. Although our study did not reveal substantial enhancements in neurological outcomes in near-term lambs following the UCO, it highlights the necessity to explore clemastine's broader therapeutic potential. Clemastine exhibits a diverse range of desirable mechanisms of action that have demonstrated neuroprotective effects both in vitro and in vivo. Clemastine has immunomodulatory properties on a systemic level[45], as well as local level by regulation of microglial responses [46], attenuation of neuronal apoptosis [46], and astrocytes loss [47]. One of the most significant effects of clemastine is increase in oligodendrocyte differentiation and enhancement of remyelination shown in various animal models [47]. Most importantly, success of clemastine as a remyelination therapy has been noted in clinical trials of relapsing multiple sclerosis with chronic demyelinating optic neuropathy[22]. In HIE, oligodendrocytes are particularly vulnerable to hypoxia and hypoglycemia [48]. Evidence suggests that in attempts to rescue demyelinated areas following a stroke, there is an increase in the density of perivascular oligodendrocyte progenitor cells [48]. In term infants in particular, myelinated fibers are more metabolically active and hence more vulnerable to HIE[49]. This vulnerability makes clemastine an ideal neurotherapeutic candidate, considering clemastine's demonstrated potential to protect and restore myelin, along with its favorable pharmacokinetic profile and minimal toxicity confirmed by our study. Thus, further investigation is warranted, ideally in a clinical setting, to explore clemastine as a neuroprotective agent for HIE.”  

Please add the chapter of conclusions

Thank you for this comment. We have now added chapter 5. Conclusions:

-p.14 Ln 498:

 “The evaluation of clemastine as a potential neuroprotective agent in neonates at risk of HIE has revealed several findings and considerations. While clemastine has shown therapeutic promise for demyelinating disorders and in several animal models of HIE, its use in neonates for neuroprotection remains an area of active investigation. Our study in neonatal lambs mimicked the pharmacokinetic parameters of young children following 2 mg/kg IV clemastine infusion that were substantially higher than those that exhibited neuroprotection in adults with MS. Clemastine’s wide receptor selectivity could lead to potential adverse effects, most importantly impaired alertness and arrhythmias. Despite UCO itself being a trigger for altered mental status and impaired hemodynamics in lieu of hypoxia, hypercarbia and acidosis, administration of clemastine did not lead to concerning side effects or end organ toxicity in our model. Clemastine may exhibit its effects by modulation of immune responses and these effects need to be understood and defined in  clinical trials in the setting of neonatal HIE. While clemastine administration did not translate into significant improvements in neurodevelopmental outcomes in our study, the improvement of select biochemical and inflammatory markers warrant comprehensive assessment of clemastine’s potential as a neurotherapeutic for HIE by further studies with longer follow-up durations, larger sample sizes, and consideration of potential gender differences.”

Reviewer 2 Report

Authors describe research on hypoxic-ischemic encephalopathy (HIE) and clemastine, a compound with antihistamine properties. The study finally reports limited neuroprotection with clemastine in a 6 days follow up period after birth. Authors fully acknowledge some of the limitations of their study such as short follow up period and small sample size.

I find that this is a well written study that appropriately describe its methodology, clearly present the results and correctly discusses their findings in light of published literature.  However, graphs should report each data point and not only average values. Moreover, even with a small sample size, data points from males and females should be colored so that the reader can judge possible sex related differences. Authors should also first analyze the data from males and females separately and report their findings. After that, the data can be pooled and reanalyzed as per author´s choice.

The impact of this study could be enhanced by incorporating a more detailed analysis of neuroinflamation using flow cytometry and molecular markers. Likewise, cytokine content could be assessed. I also find that common reflexes and additional behavioral observations could have been incorporated to better understand neurological outcomes.  

Additional questions:

What is the origin of the animals? In which conditions were their kept before and during the study?

Author Response

Reviewer 2:

Authors describe research on hypoxic-ischemic encephalopathy (HIE) and clemastine, a compound with antihistamine properties. The study finally reports limited neuroprotection with clemastine in a 6 days follow up period after birth. Authors fully acknowledge some of the limitations of their study such as short follow up period and small sample size.

I find that this is a well written study that appropriately describe its methodology, clearly present the results and correctly discusses their findings in light of published literature.

However, graphs should report each data point and not only average values. Moreover, even with a small sample size, data points from males and females should be colored so that the reader can judge possible sex related differences. Authors should also first analyze the data from males and females separately and report their findings. After that, the data can be pooled and reanalyzed as per author´s choice.

Thank you for this comment. We have edited our graphs to clearly show individual values where applicable. We have additionally  edited the graphs, analyses, and text  to reflect sex-specific differences and introduced them prior the pooled data in subsections of the manuscript.

The impact of this study could be enhanced by incorporating a more detailed analysis of neuroinflamation using flow cytometry and molecular markers. Likewise, cytokine content could be assessed.

Thank you for this suggestion. We will include the flow cytometry and cytokine analyses in our future studies. Attempts to address these issues at this juncture would require several months of additional work.

I also find that common reflexes and additional behavioral observations could have been incorporated to better understand neurological outcomes.

Thank you for this suggestion. Our neurobehavioral assessment is based on observations previously conducted in sheep to monitor their well-being following birth (Dwyer C et al. Effect of ewe and lamb genotype on gestation length, lambing ease and neonatal behaviour of lambs. Reprod Fertil Dev. 1996;8(8). DOI: 10.1071/RD9961123 and Castillo-Melendez M et al.Experimental modelling of the consequences of brief late gestation asphyxia on newborn lamb behaviour and brain structure. PLoS One. 2013;8(11). DOI: 10.1371/journal.pone.0077377, Mike et al. Defining longer term outcomes in an ovine model of moderate perinatal hypoxia-ischemia Dev Neurosci. 2022; 44(4-5): 277–294). It includes the level of alertness/encephalopathy, sucking reflex, and gross motor function that includes balance, coordination, strength and overall locomotor activity. Some of the tests are limited by the scope of the laboratory, condition of the animal and the duration of observation. For example, the Morris water maze test is not applicable in animals that are encephalopathic or cannot use their limbs.

Additional questions:

What is the origin of the animals?

Animals are white dorper sheep of both sexes delivered at near term (near-term 141-143 days gestation; term ~ 147-150 days).

In which conditions were their kept before and during the study?

The sheep are kept in conditions regulated by the Guide for the Care and Use of Agricultural Animals Used in Research (Ag Guide), USDA Animal Welfare Regulations (AWR) and the Guide for the Care and Use of Laboratory Animals (Guide). All handling for research must be approved by the Attending Veterinarian and the IACUC.

Before the study: Pregnant ewes are observed and checked for health concerns daily. Adequate feed and potable water is available. Temperature of the housing fits into the range 61°F to 81°F.

After the study: the ewe is euthanized. The lambs are kept on a radiant warmer to maintain normothermia until weaned from mechanical ventilation and extubation. After extubation, the lambs are placed in a pento recover. Lambs are assessed Q4-6hrsh by trained personnel, and tube fed within3 hours following resuscitation  and until they are able to bottle feed. For the scope of this study, we do not use hypothermia. Study participants as well as laboratory and data analysis personnel are blinded to treatment group throughout the duration of the study.

Round 2

Reviewer 2 Report

In the methods section is still not clearly mentioned from where the animals were obtained and in which conditions they were kept. These details can help other laboratories attempting to replicate the study. Where the animals grown in house or purchased from a particular vendor? in which conditions were they housed? what and how were they fed.. things like that. 

Author Response

Response to Reviewer:

In the methods section is still not clearly mentioned from where the animals were obtained and in which conditions they were kept. These details can help other laboratories attempting to replicate the study. Where the animals grown in house or purchased from a particular vendor? in which conditions were they housed? what and how were they fed.. things like that. 

Thank you for this comment. We revised the Materials and Methods section of the manuscript and edited a section “2.1 Animals” to describe in more detail the animal vendor, housing conditions, feeding regimen, observations.

P.3, Ln.95:

“2.1. Animals 

The data in this work is available upon request. All animal research was approved by the University of California Davis Institutional Animal Care and Use Committee and was performed in accordance with the Guide for the Care and Use of Laboratory Animals.  The sheep were kept in conditions regulated by the Guide for the Care and Use of Agricultural Animals Used in Research (Ag Guide), USDA Animal Welfare Regulations (AWR) and the Guide for the Care and Use of Laboratory Animals (Guide). All handling for research was approved by the Attending Veterinarian and the IACUC. White dorper sheep were purchased from Luke Vanlaningham at ~133 days gestation. The ewes were maintained in a barn where they were fed alfalfa and grass hay twice daily and had access to fresh water. The ewes were not fed the night before the UCO. After the UCO the ewes were euthanized. The lambs were kept on a radiant warmer to maintain normothermia until weaned from mechanical ventilation and extubation. After extubation, the lambs were placed in a pen to recover. Lambs were kept under carefully regulated conditions to ensure their well-being and health and to minimize potential sources of variability and stress that could impact the research findings. The housing and care of these animals strictly adhered to ethical and regulatory standards. The lambs were housed in a controlled environment where factors such as temperature, humidity, and lighting were closely monitored to provide a comfortable and stress-free setting. The lambs were subject to Q4-6h observations by trained personnel to check for any signs of health concerns or distress, allowing for early detection of any health concerns and for assessment of neurodevelopmental outcomes. Feeding schedule followed tube feeding within 3 hours following resuscitation and therafter until lambs were able to bottle feed. The lambs were given a commercially available colostrum replacer the first 24 hours and then transitioned to a commercially available milk replacer. They were housed in a 5 ft by 5ft pen in the same barn the ewes had been maintained in on rubber mats and shavings and had two heat lamps turned on during the 10pm feeding and turned off again at 8am to facilitate the milk digestion. For the scope of this study, we did not use hypothermia. Study participants as well as laboratory and data analysis personnel were blinded to treatment group throughout the duration of the study.”